# Dentin Bonding Durability of Four Different Recently Introduced Self-Etch Adhesives

**DOI:** 10.3390/ma17174296

**Published:** 2024-08-30

**Authors:** Sayaka Kitahara, Shojiro Shimizu, Tomohiro Takagaki, Masanao Inokoshi, Ahmed Abdou, Michael F. Burrow, Toru Nikaido

**Affiliations:** 1Department of Operative Dentistry, Division of Oral Functional Science and Rehabilitation, School of Dentistry, Asahi University, 1851 Hozumi, Mizuho, Gifu 501-0296, Japan; sayaka71@dent.asahi-u.ac.jp (S.K.); takagaki@dent.asahi-u.ac.jp (T.T.); nikaido-ope@dent.asahi-u.ac.jp (T.N.); 2Department of Oral Devices and Materials, Graduate School of Medical and Dental Sciences, Tokyo Medical and Dental University, 1-5-45 Yushima, Bunkyo, Tokyo 113-8549, Japan; 3Department of Restorative Dentistry, Faculty of Dentistry, University Malaya, Kuala Lumpur 50603, Malaysia; 4Faculty of Dentistry, The University of Hong Kong, Pokfulam, Hong Kong SAR, China; mfburr58@hku.hk

**Keywords:** self-etch adhesive, universal bond, micro-tensile bond strength, degree of conversion, thermal cycle

## Abstract

The objectives of this study were to evaluate the bonding durability of four different self-etch adhesives to dentin after 24 h and thermal cycling (TC) and to measure the degree of polymerization conversion (DC) in situ. Two-step self-etch adhesives, Clearfil SE Bond 2 (SE2, Kuraray Noritake Dental) and G2-Bond Universal (G2B, GC), and one-step self-etch adhesives, Scotchbond™ Universal Plus Adhesive (SBU, 3M ESPE) and Clearfil Universal Bond Quick (UBQ, Kuraray Noritake Dental), were used. The labial surfaces of bovine teeth were ground to create flat dentin surfaces. The adhesives were applied according to the manufacturers’ instructions. After resin composite buildup and 24 h water storage, the specimens were sectioned into beams and all groups were subjected to thermal stress for 0, 10,000 (10k), or 20,000 (20k) cycles followed by micro-tensile bond strength (µTBS) testing. In situ DC was investigated with a laser Raman microscope. The µTBS data were statistically analyzed and subjected to a Weibull analysis. The different groups were compared at the characteristic strength (63.2% probability of failure) (α = 0.05). Two-Way ANOVA was used to show the effect of different adhesives and thermal cycling on the mean DC% followed by Tukey’s multiple comparison post hoc test. G2B/TC10k resulted in a significant increase in the µTBS compared to TC0. SBU/TC20k showed significantly higher µTBS compared to TC0. For comparison between different tested adhesives, SBU showed a significantly lower µTBS compared to G2B after TC10k. G2B and SBU showed a greater number of adhesive failures after TC. Mean DC% was different for each adhesive. The newly developed MDP- and HEMA-free 2-SEA showed similar bonding performance with the gold-standard 2-SEA. However, there is still room for further improvement in terms of SEAs.

## 1. Introduction

Self-etch adhesives with one step (1-SEA) or two steps (2-SEA) were introduced to overcome the problems of etch and rinse systems and have been proven to exhibit good bonding performance and sealing ability [1,2]. 1-SEAs are quite well accepted clinically due to their simple clinical application and time saving application. Although 1-SEAs are simplified in the method and composition, they did not promote effective dentin bonding performance when first introduced [3]. Thereafter, manufacturers have modified the chemical composition of 1-SEAs, resulting in improved bonding performance [4]. Recently, so-called “universal adhesives”, which contain different functional monomers according to manufacturers, have been released. These adhesives aimed to be simple and versatile, claiming to be able to pre-treat both tooth and restoration surfaces in conjunction with different etching modes. However, their ability to achieve adequate and durable bonding performance as a “universal adhesive” appears to be limited [5,6].

Being not only for universal usage, 1-SEAs have also been modified in some other respects. For example, 1-SEAs tend to show higher water sorption compared with 2-SEAs due to their hydrophilic character, which then exhibits a greater reduction in the mechanical properties of the adhesive layer over time [7]. Thus, manufacturers tried to reduce or replace 2-hydroxyethyl methacrylate (HEMA), which has a hydrophilic nature even after polymerization. However, HEMA-free 1-SEAs had some drawbacks such as phase separation [8] and reduced wettability on dentin [9]. This led some manufacturers to launch an amide monomer-containing 1-SEA. The amide monomer exhibits an extremely hydrophilic character compared with HEMA before curing, but converts to become a hydrophobic polymer after polymerization [7]. Furthermore, some 1-SEAs claiming a shorter or 0 s application time have been launched and reported to be successful in initial bonding performance [7]; however, they have not shown durable bonding after thermal stress or water storage [10].

Both in 1-SEAs and 2-SEAs, functional monomers are the essential components to achieve their clinical efficiency. Additionally, these functional monomers play a primary role in “universal” usage due to the potential chemical interaction to metal oxides [11]. The functional monomers comprise both hydrophobic and hydrophilic moieties, which enhances monomer penetration and provides a potential chemical interaction to the bonding substrates [12]. Among these functional monomers, 10-methacryloyloxydecyl dihydrogen phosphate (MDP) is capable of forming strong ionic bonds with calcium, and forming a salt that has a relatively low solubility [13]. Indeed, adhesives containing MDP have shown consistently reliable long-term adhesive performance in numerous laboratory and clinical studies [14]. In two previous studies [15,16], MDP in the bonding agent of 2-SEAs led to a rise in water sorption in the adhesives after curing, which affected the mechanical properties of the adhesive layer. With this in perspective, a manufacturer launched a new 2-SEA, which claimed that it did not contain HEMA or MDP in the bonding agent (G2-Bond Universal, G2B, GC, Tokyo, Japan). The primer of G2B contains a similar component to a HEMA-free 1-SEA (G-Premio Bond, GC). On the other hand, the bonding agent of G2B consists of hydrophobic monomers, to ensure its long-term mechanical stability against water sorption [17]. 1-SEA Universal adhesives contain MDP and HEMA that may suffer hydrolytic degradation due to inadequate storing conditions and long storage times [18,19]. It seems that the components of universal adhesives affect an important role in their stability. On the other hand, G2B is designed to exclude the effects of storage conditions and is further structured into a 2-SEA. However, there are few laboratory studies comparing the bonding durability of those recently introduced 1-SEAs and 2-SEAs. Thermal cycling simulates the effect of varying temperatures in the oral cavity. For that reason, thermal cycling is usually performed between the temperature of 5 and 55 °C, each dwell time of 30 s, and the transfer time of 5 s. It is proposed that 10,000 cycles might represent 1 year of service [20,21].

The objectives of this study were (1) to evaluate the micro-tensile bond strength (μTBS) of two contemporary universal adhesives, a contemporary 2-SEA and a novel universal adhesive 2-SEA, to bovine dentin after 24 h and after thermal cycling (TC); (2) to measure the degree of conversion (DC) in situ after 24 h and after TC of the adhesives; and (3) to determine whether a correlation between μTBS and DC before and after TC exists. The null hypotheses tested in this study were (1) there are no differences in μTBS before and after TC, and (2) there are no differences in DC of the adhesive layer before and after TC.

## 2. Materials and Methods

### 2.1. Materials

Table 1 provides the detailed description of the respective adhesives and the application protocols. Four adhesives, Clearfil SE Bond 2 (SE2; Kuraray Noritake Dental, Tokyo, Japan), G2-Bond Universal (G2B; GC, Tokyo, Japan), Scotchbond™ Universal Plus Adhesive (SBU; 3M ESPE, St. Paul, MN, USA), and Clearfil Universal Bond Quick ER (UBQ; Kuraray Noritake Dental), were used in this study.

### 2.2. Micro-Tensile Bond Strength Test

The sample preparation protocol is shown in Figure 1.

Recently extracted bovine incisors, stored frozen, were used as the bonding substrate. The labial surfaces of the teeth were ground using a model trimmer under water lubrication to expose areas of middle dentin; the roots were cut off and the pulpal tissue removed. The exposed dentin surfaces were wet-ground with 600-grit SiC paper (Sankyo Rikagaku, Saitama, Japan) to create flat surfaces with a standard smear layer [22].

Each tooth was bonded according to the manufacturers’ instructions (Table 1).

The primers of 2-SEAs were applied in 20 s for SE2 and 10 s for G2B. The air blowing time and air pressure were 5 s for SE2 with gentle air blowing and 5 s for G2B with air blowing at maximum air pressure to dry. The bonds of 2-SEAs were further applied and a uniform bond film was formed by gentle air blowing. The adhesives of 1-SEAs were applied in 20 s for SBU and applied bond-wise with a rubbed motion (no waited time) for UBQ. Both adhesives air dry for at least 5 s until the adhesive does not move anymore.

After, the specimens were cured with a light-emitting diode (LED) light-curing unit (Pencure 2000, normal mode at 1100 mW/cm^2^, Morita, Osaka, JAPAN) for 10 s. Then, a 6 mm thick layer of the resin composite was placed in three increments (Clearfil AP-X, shade A2, Kuraray Noritake Dental) and each increment was light-cured for 20 s using the light-curing unit. After storage in 37 °C distilled water for 24 h, all bonded specimens were sectioned into beam-shaped specimens (surface area: 1.0 × 1.0 mm) using a low-speed diamond saw (Isomet, Buehler, Lake Bluff, IL, USA).

The beam-shaped specimens in each group were randomly divided into three subgroups, and exposed to either 0, 10,000, or 20,000 thermocycles (TCs). The beams were cycled between two water baths at 5 and 55 °C with a dwell time of 30 s in each bath and a transfer time of 5 s using a thermocycling device (K178-08, Tokyo Giken, Tokyo, Japan). After the TC, each specimen was individually bonded to a tensile testing jig using a cyanoacrylate adhesive (Zapit, Dental Ventures of American, Anaheim Hills, CA, USA) mounted in a tabletop testing machine (EZ-SX, Shimadzu, Kyoto, Japan) and subjected to the μTBS test at a crosshead speed of 1.0 mm/min (Figure 1). The preparation of adhesive samples was carried out at 23 ± 1 °C and a relative humidity of 50 ± 5%, and measurements.

After the μTBS test, the fractured specimens were osmium-coated (HP-IS, Vacuum Devices, Mito, Japan). Moreover, they were observed using a scanning secondary electron image electron microscope (SEM, S-4500, Hitachi, Tokyo, Japan) with an accelerating voltage of 15 kV under 70× magnification to characterize the failure modes. Fractured specimens were classified into one of three categories—adhesive failure (A): if 80–100% of the failure occurred at the adhesive–substrate interface, mixed failure (M): if at least two of the aforementioned failure patterns were observed but on less than 80% of the fracture surface, or cohesive failure (C): the cohesive in the adhesive layer or dentin, and if 80–100% of the failure occurred within the adhesive layer or in the underlying dentin [17].

### 2.3. Measurement of In Situ Degree of Conversion (DC%)

Twelve bovine dentin specimens (*n* = 3 per adhesive) were randomly assigned for each group as described earlier for the μTBS test. The 2-SEAs as primers and adhesives or 1-SEAs as adhesives were applied to the dentin surface, and composite resin buildups were constructed on the bonded dentin in the same manner described for the μTBS test. After storage of the bonded specimens in distilled water at 37 °C for 24 h, the resin–dentin specimens were horizontally sectioned across the bonded interface with a low-speed diamond saw (Isomet) to expose the adhesive–dentin interface. The specimens in each group were randomly divided into three subgroups, and exposed to either 0, 10,000, or 20,000 TC.

An in Via Raman microscope (in Via Reflex—INVIA0617-03; Renishaw PLC, New Mills, Wotton-under-Edge, Gloucestershire, UK) using monochromatic radiation emitted by a semiconductor laser output (wavelength of 785 nm and excitation power of 100 mW: TC0 or 10 mW: TC10k and TC20k) was employed as the one applied on target. TC10k and TC20k were set to 10 mW to avoid charring the samples. For the Raman analysis, the magnification of the lens used was 100×. It was equipped with a confocal microscope, and a piezoelectric XY stage with a minimum step width of 1 µm, and the focus of the laser beam in conjunction with the CCD camera provided a spatial resolution of 1 µm. Spectra were taken at the dentin–adhesive interface from a 1 to 3 µm distance from the dentin site. Five sites were examined per dentine–adhesive slice.

Spectra of uncured adhesives were taken as a reference. Post-processing of spectra was performed using Fityk software (version 0.9.8). The ratio of double-bond content of the monomer to polymer in the adhesive was calculated according to the following formula:DC (%) = (1 − R_(cured)_/R_(uncured)_) × 100(1)
where “R” is based on the spectrum of the benzene ring derived from the monomer contained in the resin (1609 cm^−1^); the ratio of the spectrum of the double bond (C=C) of the unpolymerized monomer (1639 cm^−1^) was calculated.

### 2.4. Statistical Analysis

Normality of the residuals was checked using the Shapiro–Wilk test and showed a normal distribution of the TBS and DC% values. The µTBS data were statistically analyzed using a Weibull analysis (R4, R: a language and environment for statistical computing. R Foundation for Statistical Computing, Vienna, Austria). Weibull parameters were calculated by Wald estimation, and pivotal confidence bounds were calculated with Monte Carlo simulation. The different groups were compared at the characteristic strength (63.2% probability of failure) (α = 0.05). Two-Way ANOVA was used to show the effect of different adhesives and thermal cycling on the mean DC% followed by Tukey’s post hoc test for multiple comparisons using statistical software (SPSS ver. 26.0 for Windows, IBM Corp., Armonk, NY, USA). The sample size was estimated using a G*power 3.1.9.7 (Düsseldorf, Germany) analysis based on a preliminary study. α error = 0.05, 1-type II error = 0.8, mean differences = 15.7, SD = 8.7, effect size = 0.8, minimum number of specimens = 29.

## 3. Results

Results of the µTBSs are presented in Table 2.

For G2B, TC10k resulted in a significant increase in characteristic strength compared to TC0 (*p* < 0.05). However, increasing the number of the TC from 10k to 20k resulted in a decrease in the characteristic strength, which was insignificantly different from TC0 and TC10k (*p* < 0.05). For SE2, TC did not affect the results of the µTBSs. However, the characteristic strength of SE2 was not significantly different between TC0 and TC20k (*p* > 0.05). For SBU, TC20k showed significantly higher characteristic strength compared to TC0 (*p* < 0.05). Meanwhile, for UBQ (a 1-SEA), the characteristic strength was not significant among all TC groups. For comparison between the different tested adhesives, SBU showed the only significantly lower characteristic strength compared with G2B after TC10k. For all adhesives, a not significant difference in the characteristic strengths was observed between each adhesive within the corresponding TC group. In the four tested adhesives, SBU showed a significantly lower characteristic strength compared with G2B after TC10k.

A Weibull analysis probability plot for tested groups is represented in Table 2, and Figure 2 and Figure 3.

The difference found in characteristic strength among four adhesives for each TC group was not significant (*p* > 0.05). However, SE2 showed the lowest Weibull modulus among all the adhesives at TC0; however, a not significant difference at TC10k and TC20k was determined. For SBU and UBQ, there was a tendency for the slope to decrease for TC10k and TC20k, indicating that the variation in bond strength increases depending on TC, but TC10k and TC20k resulted in a significant decrease in the Weibull modulus for UBQ only. For UBQ, TC10k and TC20k resulted in a significant decrease in the Weibull modulus.

The failure mode analysis is presented in Table 2, accompanied by representative SEM failure mode images depicted in Figure 4.

G2B and SBU tended to show a greater number of adhesive failures (A) after TC, whereas SE2 and UBQ showed a decreased number of adhesive failures (A) and increased cohesive failures (C).

Results of the DC% are presented in Table 3 and Figure 5.

Two-Way ANOVA showed that different adhesives resulted in a significant effect on mean DC% at *p* < 0.001. Meanwhile, thermal cycling resulted in no significant effect at *p* = 0.611. At TC0, a significantly higher DC was observed for SBU and UBQ compared with G2B and SE2 (*p* < 0.001). For TC10k and TC20k, all adhesives showed a significant difference between each other (*p* < 0.001). No significant difference in DC% was observed in SE2 *(p* = 0.484) and SBU (*p* = 0.720) before and after TC. G2B and TC20k showed significantly lower DC% compared to TC0 and TC10k (*p* < 0.001). For UBQ, TC20k showed a significantly lower DC% compared to TC0 only (*p* = 0.003).

## 4. Discussion

In this study, the dentin bonding performance of two recently introduced 1-SEA and one 2-SEA universal adhesive was compared with SE Bond 2, which is the gold-standard two-step SEA. The null hypotheses tested in this study were (1) there are no differences in μTBS before and after TC; (2) there are no differences in DC of the adhesive layer before and after TC. They were both partially rejected.

The bonded specimens were cut into 1 mm × 1 mm beam-shape specimens before conducting the TC test in this study to avoid the surrounding tooth and composite protecting the bonded interface from degradation. In order to analyze the dentin bond durability of adhesives, thermal cycling for 10,000 and 20,000 times was chosen instead of long-term water storage. Long-term water storage may be one of the ideal and simplified aging methods [18,19]. However, it takes a long time to evaluate the results obtained. Thermal cycling caused contraction/expansion stresses and accelerated chemical degradation at the interface. In the tiny µTBS specimens, the artificial aging effect induced by TC can occur due to hydrolysis of the interface components, and subsequent uptake of water and the extraction of breakdown products or poorly polymerized resins [23].

A Weibull analysis was employed to statistically analyze the adhesion test results. A Weibull analysis has become prevalent in materials testing in recent years, particularly in destructive testing where results often do not conform to a normal distribution but instead follow the Weibull distribution, which is based on the weakest link model. In a Weibull analysis, the cumulative failure probability is plotted on the vertical axis, and the material strength (micro-tensile adhesive strength in this experiment) is plotted on the horizontal axis. This allows for the approximation of straight lines for each group. Comparing these straight lines across different groups enables the recognition of characteristics that may not be apparent from just the average value or standard deviation obtained in an adhesion test.

For G2B, TC10k resulted in a significant increase in the characteristic strength compared to TC0. However, TC20k resulted in a significant decrease in the characteristic strength compared to TC10k. On the other hand, SE2 did not show any characteristic strength among TC0, TC10k, and TC20k.

In regard to the failure mode analysis, SE2 showed fewer adhesive failures and more cohesive failures after TC. On the other hand, G2B showed an increased number of adhesive failures after TC10k and TC20k. These differences may be associated with differences in post-cure polymerization behaviors in the two 2-SEAs. The heat challenge in TC may accelerate the post-cure polymerization of the adhesive resin, increasing the DC of the adhesives. This is an advantage to increase the bond strength values of both SE2 and G2B.

HEMA has a high hydrophilic monomer, which is prone to hydrolytic degradation over time. A G2B primer is a HEMA-free and moderately acidic primer (pH = 1.5) [24]. This is a good advantage in G2B to prevent dentin bond durability [8,9]. For G2B, the primer and the bond contain fillers, which may enhance the mechanical property of the bond layer.

SE2 contains MDP both in the primer and the bond, while G2B contains it only in the primer. MDP has a hydrophobic and hydrophilic structure, which may exhibit the characteristics of water sorption after polymerization. It was suggested that the absence of MDP might reduce permeability into dentin [15]. Such factors influenced the results in the micro-tensile bond strength testing and fracture mode analysis in the 2-SEAs.

Except for the hydrophilic mono-functional monomers, such as HEMA and MDP, a higher degree of the conversion and mechanical property in the bonding layer was contributed to [25], but it resulted in less penetration into the demineralized underlying dentin substrate [13,15]. It is difficult to conclude whether HEMA of 2-SEA should be included; compensation for the loss of dentin permeability and reduction in water permeability is still a big problem. For the 1-SEA of SBU, TC20k showed a significantly higher characteristic strength than TC0, while for UBQ, the characteristic strength was not significant for all the TC groups. SBU showed improved bond durability compared with the previous product, Scotchbond Universal (3M ESPE) [6], which might be due to its optimized amount of HEMA and newly adapted crosslinking BPA-free resin. HEMA content in the adhesive was reported to increase the amount of water in the adhesive layer and to decrease bond durability [26]. UBQ contains an amide monomer, showing quite high hydrophilicity before curing, thus being capable of the reduction of HEMA [7]. Therefore, UBQ showed a reduction in adhesive failure according to the number of TCs, which may be due to the prevention of degradation at the dentin–bonding interface. Among all the four adhesives of TC20k, no significant differences were found in dentin bond strength. The bonding performance of 1-SEAs improved over the past decade, and the previous studies indicated that the recent 1-SEAs are closely comparable to 2-SEAs [27].

Micro-Raman spectroscopy is a useful analytical technique for accurately determining the degree of conversion of a small area. Compared with other methods such as FT-IR ATR, micro-Raman spectroscopy is able to analyze a smaller area of interest, like a bonding layer in the sectioned bonded interface. Many previous studies reported the DC of the bonding layer at the bonded interface with the use of Raman microscopy; nowadays, this type of method is a so-called “in situ DC” [28,29,30,31]. A previous study demonstrated that the DC of the adhesive layer is dependent on the location, and tends to be lower in the oxygen inhibition layer near the resin composite [32]. Thus, spectra were taken at the dentin–adhesive interface, located a 1 µm distance from the dentin site in 1-SEAs and 3 µm distance in 2-SEAs to avoid inequitable results.

The bonding resin of G2B is HEMA-free and MDP-free, resulting in a hydrophobic character. The hydrophobicity of the bonding layer with less water sorption is favorable for less degradation by water uptake. However, G2B exhibited relatively low DC, which decreased after TC. The hydrophobic character of the G2B bonding agent had an effect on the penetration and curing in the presence of water and collagen in dentin, which are both hydrophilic in nature [15,33]. It might be due to that in situ DCs were recorded adjacent to dentin like a 3 µm distance from the dentin site. The previous studies on 1-SEA [34,35,36] suggested that differences in DC% in 1-SEAs would likely be influenced by several factors, such as the type and amount of solvents, monomers, and initiators used. The changes in DC% were observed with TC in the current study. Different viscosity of each bonding resin may influence the rate of the diffusion reaction into the underlying dentin. However, the phenomenon was still unclear. Further research should be carried out.

In a previous study [32], in situ DC of Clearfil SE Bond (Kuraray Noritake Dental) was reported to be 78.5–88.1%, which is similar to our result in SE2. However, in another study [30], UBQ exhibited 59.9% and Scotchbond Universal showed 59.2% as the mean DCs in situ in a similar experimental design. These lower DCs could be due to a different wavelength of the diode laser in Raman microscopy, in which the 785 nm diode laser tended to show higher DC% compared with a 532 nm diode laser. Therefore, it would be difficult to compare directly with the previous reports. UBQ showed a higher mean in situ DC compared with the other three adhesives tested. Tichy et al. [37] investigated DC values of UBQ and UBQ without the amide monomer, and showed a lower DC value in UBQ without the amide monomer group. The improved DC might be attributed to the lower hydrophilicity of UBQ compared to UBQ without the amide monomer. SBU showed relatively lower DC compared with UBQ. The amide monomer has a hydrophilic character; HEMA is relatively reduced in UBQ. SBU showed relatively lower DC in the tested 1-SEAs possibly due to its higher content of HEMA. In this study, DC of SBU showed a trend similar to previous reports [38].

The findings of the current study suggest that recently introduced so-called “universal adhesives” demonstrate comparable bonding performance to dentin with gold-standard 2-SEA SE Bond 2. However, more clinical bonding performance like simulated pulpal pressure or in situ clinical evaluation should be carried out in future studies.

## 5. Conclusions

The results of this study indicated that no significant differences in micro-tensile bond strength were observed among the four adhesives after thermal cycling. Therefore, the more recent 1-SEAs or newly developed MDP- and HEMA-free 2-SEA showed similar bonding performance with the gold-standard 2-SEA. However, the fracture surface analysis revealed that the proportion of interfacial failure between the adhesive and dentin increased after thermal cycling in G2B and SBU. Changes in the degree of conversion of the bonding layer after thermal cycling differed among the materials; the degree of conversion decreased in G2B, while it increased in UBQ.

There is still room for further improvement in terms of the chemical composition and system of SEAs to achieve durable bonding.

## Figures and Tables

**Figure 1 materials-17-04296-f001:**
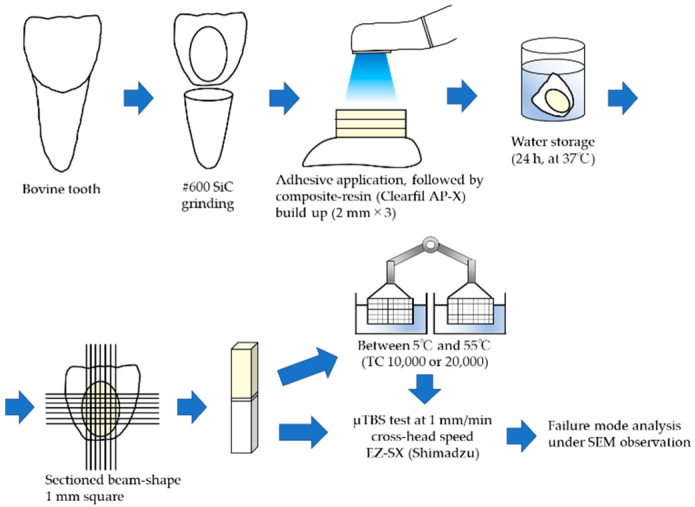
Schematic illustration of methodology of specimen preparation for μTBS.

**Figure 2 materials-17-04296-f002:**
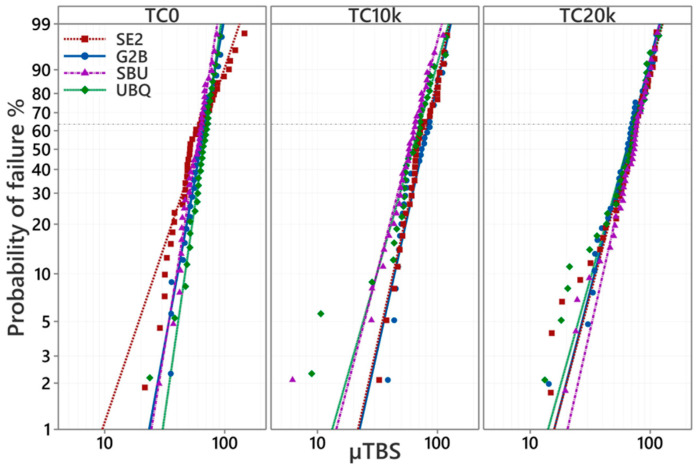
The Weibull probability plot of the µTBS (MPa) of four adhesives in each TC. T0: No thermal cycles, TC10k: 10,000 thermal cycles, TC20k: 20,000 thermal cycles.

**Figure 3 materials-17-04296-f003:**
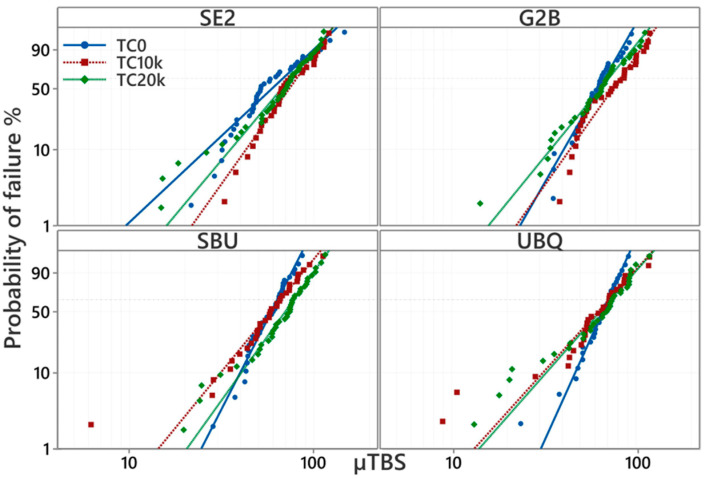
The Weibull probability plot of the µTBS (MPa) of each TC in four adhesives. T0: No thermal cycles, TC10k: 10,000 thermal cycles, TC20k: 20,000 thermal cycles.

**Figure 4 materials-17-04296-f004:**
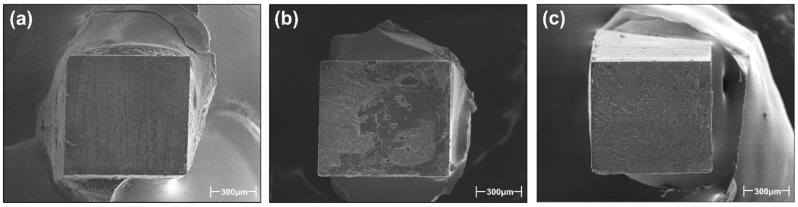
Representative SEM images of failure mode at magnification of 70×. (**a**) A: adhesive failure, (**b**) M: mixed failure, (**c**) C: cohesive failure.

**Figure 5 materials-17-04296-f005:**
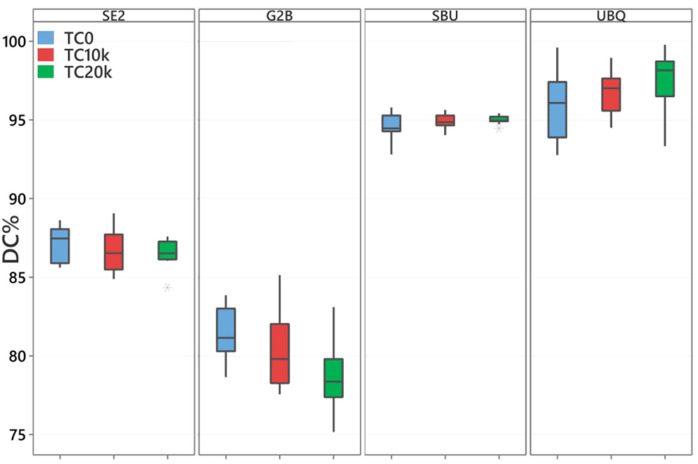
The box-plot, median, and standard deviation of the values of DC% in the different adhesive systems and TC.

**Table 1 materials-17-04296-t001:** The composition of the materials used in this study and their application procedures.

Code	Materials	Components	Manufacturer	Application Procedure
Adhesive				
SE2	CLEARFIL SE Bond 2	Primer: 10-MDP, water, HEMA, hydrophilic dimethacrylate, CQ, N,N-diethanol p-toluidine Bond: 10-MDP, Bis-GMA, HEMA, hydrophobic dimethacrylate, CQ, N,N-diethanol p-toluidine, silanated filler	Kuraray Noritake Dental, Tokyo, Japan	1. Apply the primer and leave for 20 s. 2. Gentle air blowing for 5 s. 3. Apply the bond. 4. Make a uniform bond film using a gentle air flow. 5. Light cure for 10 s.
G2B	G2-BOND Universal	1-Primer: water, 4-MET, 10-MDP, MDTP, dimethacrylate, acetone, filler, photoinitiator 2-Bond: dimethacrylate, filler, photoinitiator	GC, Tokyo, Japan	1. Apply 1-Primer and leave for 10 s. 2. Dry thoroughly with air under maximum air pressure for 5 s. 3. Apply 2-Bond. 4. Gentle air blowing to make the film uniform. 5. Light cure for 10 s.
SBU	Scotchbond™ Universal Plus Adhesive	10-MDP, HEMA, vitrebond copolymer, dimethacrylate resins (BPA derivative-free), ethanol, water, initiators, dual-cure accelerator, optimized mixture of silane, filler	3M ESPE, St. Paul, MN, USA	1. Apply with agitation for 20 s. 2. Air dry for at least 5 s until adhesive does not move anymore. 3. Light cure for 10 s.
UBQ	CLEARFIL Universal Bond Quick ER	10-MDP, Bis-GMA, HEMA, hydrophilic amide monomer, colloidal silica, ethanol, DL-camphorquinone, accelerators, water, sodium fluoride	Kuraray Noritake Dental, Tokyo, Japan	1. Apply bond with a rubbing motion (no waiting time). 2. Dry by blowing mild air until bond does not move (5 s). Use a vacuum aspirator to prevent bond from scattering. 3. Light cure for 10 s.
**Resin composite**				
	CLEARFIL AP-X	Bis-GMA, TEGDMA, silanated baium glass filler, silanated silicafiller, silanated colloidal silica, camphorquinone, initiators, accelerators, pigments	Kuraray Noritake Dental, Tokyo, Japan	Six-millimeter-thick resin composite block was placed in three increments and each increment was light-cured for 20 s.

10-MDP: 10-methacryloyloxydecyl dihydrogen phosphate; HEMA: 2-hydroxyethyl methacrylate; Bis-GMA: bisphenol-adiglycidyl methacrylate; CQ: camphorquinone; 4-MET: 4-methacryloyloxy ethyl trimellitic acid; MDTP: 10-methacryloyloxydecyl dihydrogen thiophosphate; TEGDMA: triethyleneglycol dimethacrylate.

**Table 2 materials-17-04296-t002:** The results of the Weibull analysis and failure mode distribution.

Adh	TC	N	α [95% CI] (MPa)	β [95% CI]	FM [A/M/C]
SE2	TC0	37	69.2 [59.7 to 80.2] ^abc^	2.3 [1.9 to 3.0]	[18.9/48.6/32.4]
SE2	TC10k	33	81.7 [73.7 to 90.7] ^ab^	3.5 [2.8 to 4.7]	[15.2/60.6/24.2]
SE2	TC20k	40	75.7 [67.8 to 84.5] ^abc^	3.0 [2.4 to 4.0]	[5.0/32.5/62.5]
G2B	TC0	30	68.3 [62.6 to 74.6] ^bc^	4.3 [3.4 to 5.9]	[10.0/26.7/63.3]
G2B	TC10k	33	83.4 [75.2 to 92.5] ^a^	3.5 [2.8 to 4.8]	[39.3/50.0/10.7]
G2B	TC20k	35	72.3 [64.4 to 81.1] ^abc^	3.0 [2.4 to 4.1]	[23.5/55.9/20.6]
SBU	TC0	35	63.4 [59 to 68.2] ^c^	4.8 [3.9 to 6.5]	[14.3/46.4/39.3]
SBU	TC10k	33	65.8 [58.5 to 74.1] ^bc^	3.0 [2.4 to 4.1]	[36.4/33.3/30.3]
SBU	TC20k	39	77.7 [70.7 to 85.5] ^ab^	3.5 [2.8 to 4.6]	[20.5/41.0/38.5]
UBQ	TC0	32	70.6 [66.1 to 75.5] ^abc^	5.5 [4.3 to 7.6]	[45.2/41.9/12.9]
UBQ	TC10k	30	71.3 [62.2 to 81.8] ^abc^	2.7 [2.1 to 3.8]	[33.3/36.7/30.0]
UBQ	TC20k	33	72.9 [64.2 to 82.8] ^abc^	2.8 [2.2 to 3.9]	[18.2/30.3/51.5]

Different superscript lowercase letters within the α column are statistically significant differences based on a 95% confidence interval (CI). α: characteristic strength or scale of a Weibull parameter (MPa). β: the shape, slope, and modulus of a Weibull parameter. T0: no thermal cycles, TC10k: 10,000 thermal cycles, TC20k: 20,000 thermal cycles, FM: failure mode, A: adhesive failure, M: mixed failure, C: cohesive failure.

**Table 3 materials-17-04296-t003:** DC% in the different adhesive systems and TC.

Groups	TC0	TC10k	TC20k	*p*-Value
G2B	81.3 ± 1.6 ^aC^	80.2 ± 2.3 ^aD^	78.6 ± 2.1 ^bD^	<0.001
SE2	87.2 ± 1.1 ^aB^	86.6 ± 1.2 ^aC^	86.6 ± 0.8 ^aC^	0.484
SBU	94.6 ± 0.7 ^aA^	94.9 ± 0.4 ^aB^	95 ± 0.2 ^aB^	0.72
UBQ	95.9 ± 2.0 ^aA^	96.8 ± 1.3 ^abA^	97.7 ± 1.6 ^bA^	0.003
*p*-value	<0.001	<0.001	<0.001	-

Different uppercase letter indicates significant difference within each column (between adhesives). Different lowercase letter indicates significant difference within each row (between TC).

## Data Availability

The original contributions presented in the study are included in the article, further inquiries can be directed to the corresponding author.

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
