# Peer review of "Dentin Bonding Durability of Four Different Recently Introduced Self-Etch Adhesives"

_materials, 2024, doi:10.3390/ma17174296_

Round 1

Reviewer 1 Report

Comments and Suggestions for Authors Title: - Please eliminate "title"   Abstract: - The results are poor presented - a conclusion should be added - a clinical significance should be added   Introduction: - the originality is poor, and not clear, several paper talked about the same context - TC should be added to the aim of the study at the end of the introduction - More information could be added on the chemical composition and the application modalities of the self etch adhesives to introduce their characteristics The authors could use the following references: Iliev G, Hardan L, Kassis C, Bourgi R, Cuevas-Suárez CE, Lukomska-Szymanska M, Mancino D, Haikel Y, Kharouf N. Shelf Life and Storage Conditions of Universal Adhesives: A Literature Review. Polymers (Basel). 2021 Aug 13;13(16):2708. 
Hardan L, Bourgi R, Kharouf N, Mancino D, Zarow M, Jakubowicz N, Haikel Y, Cuevas-Suárez CE. Bond Strength of Universal Adhesives to Dentin: A Systematic Review and Meta-Analysis. Polymers (Basel). 2021 Mar 7;13(5):814. 
  Methods: - L99 why 600? add a reference - The methods for the tensile strength is poor, more details should be added - the failure mode test should be more clearly presented and the preparation for SEM should be in detail - any sample size test?   Results: - good presentation - Higher SEM images resolutions should be added - When the authors talked about adhesive failures, is between which surfaces? - any dentin cohesive failure was observed   Discussion: - The limitations are not clear - when i read the discussion, i see that all your results are presented previously by previous studies, i cannot find the originality of the present work   References   There is no reference from 2023 and 2024

Author Response

Title: - Please eliminate "title"  

We were deleted “title”

Abstract: - The results are poor presented - a conclusion should be added - a clinical significance should be added  

G2B and SBU tended to show a greater number of adhesive failures after TC. Two-Way ANOVA showed a significant effect of different adhesives on mean DC% (p<0.001). The newly developed G2B showed similar bonding performance with the gold-standard SE2. However, there is still room for further improvement in terms of technique sensitivity.

Introduction: - the originality is poor, and not clear, several paper talked about the same context - TC should be added to the aim of the study at the end of the introduction - More information could be added on the chemical composition and the application modalities of the self etch adhesives to introduce their characteristics The authors could use the following references:

Iliev G, Hardan L, Kassis C, Bourgi R, Cuevas-Suárez CE, Lukomska-Szymanska M, Mancino D, Haikel Y, Kharouf N. Shelf Life and Storage Conditions of Universal Adhesives: A Literature Review. Polymers (Basel). 2021 Aug 13;13(16):2708.

Hardan L, Bourgi R, Kharouf N, Mancino D, Zarow M, Jakubowicz N, Haikel Y, Cuevas-Suárez CE. Bond Strength of Universal Adhesives to Dentin: A Systematic Review and Meta-Analysis. Polymers (Basel). 2021 Mar 7;13(5):814.

In added to introduction of text

“1-SEA Universal adhesives contain MDP and HEMA that may suffer hydrolytic degradation due to inadequate storing conditions and long storage times [18]. It seems that the components of universal adhesives effect an important role in their stability. On the other hand, G2B is designed to exclude the effects of storage conditions and is further structured into a 2-SEA”. However, there are few laboratory studies comparing the bonding durability of those recently introduced 1-SEAs and 2-SEAs. “Thermal cycling simulates the effect of varying temper-atures in the oral cavity. For that reason, thermal cycling is usually performed between temperature 5-55℃, each dwell time 30 s, and the transfer time 5 s. It is proposed that 10,000 cycles might represent 1 year of service [19].”

 Methods: - L99 why 600? add a reference –

The grind using SiC #600 under water was assumed to be on the dentin surface cut with a steel bar. We also considered the influence of the smear layer with reference to the following paper rights.

Saikaew P, Senawongse P, Chowdhury AA, Sano H, Harnirattisai C. Effect of smear layer and surface roughness on resin-dentin bond strength of self-etching adhesives. Dent Mater J. 2018 Nov 30;37(6):973-980

The methods for the tensile strength is poor, more details should be added - the failure mode test should be more clearly presented and the preparation for SEM should be in detail - any sample size test?  

In added to text,

“The primer of 2-SEAs were applied in 20 s for SE2 and 10 s for G2B. The air blow-ing time and air pressure were 5 s for SE2 with gentle air blowing and 5 s for G2B with air blowing at maximum air pressure to dry. The bond of 2-SEAs were further ap-plied and a uniform bond film was formed by gentle air blowing. The adhesives of 1-SEAs were applied in 20 s for SBU and applied bond with a rubbed motion (no wait-ed time) for UBQ. Both adhesives air dry for at least 5 s until adhesive does not move anymore.”

“After the μTBS test, the fractured specimens were osmium-coated (HP-IS, Vac-uum Devices, Mito, Japan). Moreover, were observed using a scanning electron micro-scope (SEM, S-4500, Hitachi, Tokyo, Japan) with an accelerating voltage of 15kV under 70×magnification to characterize the failure modes. “

“Fractured specimens were classified into one of three categories; adhesive failure (A): if 80–100% of the failure occurred at the adhesive-substrate interface, mixed failure (M): if at least two of the a forementioned failure patterns were observed but on less than 80% of the fracture surface. or cohesive failure (C): cohesive in the resin composite or dentin, if 80–100% of the failure occurred within the adhesive layer or in the underlying dentin.”

The sample size was estimated using G*power3.1.9.7 analysis based on a preliminary study.

Results: - good presentation - Higher SEM images resolutions should be added - When the authors talked about adhesive failures, is between which surfaces? - any dentin cohesive failure was observed  

In added to text,

“Fractured specimens were classified into one of three categories; adhesive failure (A): if 80–100% of the failure occurred at the adhesive-substrate interface, mixed failure (M): if at least two of the a forementioned failure patterns were observed but on less than 80% of the fracture surface. or cohesive failure (C): cohesive in the resin composite or dentin, if 80–100% of the failure occurred within the adhesive layer or in the underlying dentin.”

Discussion: - The limitations are not clear - when i read the discussion, i see that all your results are presented previously by previous studies, i cannot find the originality of the present work   References   There is no reference from 2023 and 2024

In added to some new references of 2023 and 2024.

Reviewer 2 Report

Comments and Suggestions for Authors

The study is interesting, clinically useful, and very well reported. I found only two minor flaws:

1. Authors have to report what kind of international standards or guidelines (ISO or ADA or other) were use for defining number of specimens, shape and dimensions of specimens and technical details of performed laboratory tests (micro tensile bond strength test and measurement of in situ degree of conversion)?

2. Authors cited a few very old articles. Please do not cite articles older than 10 years because their are outdated in most cases. Authors can consider the latest articles strongly related to the topic e.g. doi:10.17219/dmp/133404, doi:10.17219/dmp/133071

Author Response

  1. Authors have to report what kind of international standards or guidelines (ISO or ADA or other) were use for defining number of specimens, shape and dimensions of specimens and technical details of performed laboratory tests (micro tensile bond strength test and measurement of in situ degree of conversion)?
  2.  

As you pointed out, this was done based on the ISO standard. The experimental conditions and the contents of the ISO standard are described in the text.

  1. Authors cited a few very old articles. Please do not cite articles older than 10 years because their are outdated in most cases. Authors can consider the latest articles strongly related to the topic e.g. doi:10.17219/dmp/133404, doi:10.17219/dmp/13307

As you pointed out, some of the cited references have been changed to more recent ones.

Reviewer 3 Report

Comments and Suggestions for Authors

Introduction. This investigation is a paper that presents information for researchers and clinicians in the field of adhesives in restorative dentistry. Self-etch adhesives with one-step or two-steps were introduced to overcome the problems of etch and rinse systems and well accepted clinically due to their simple clinical application and time saving application. The objectives of this study were to evaluate the micro-tensile bond strength of several contemporary universal adhesives.

This section is correct.

Materials and methods. Globally, this section is correct with several subsections (Micro tensile bond strength test; Measurement of in situ degree of conversion (DC%), Statistical analysis) that showed the steps of the experiment. Each subsection must be report a number (2.1.; 2.2; …) Also, this section must report update references of each procedure.

Results. This section is correct. The results are related with the specific characteristics of every adhesive. There are diferences  (i.e. tensile bond strength) in the different adhesives.

Discussion. This section analyzes the most important results of the study and be compared with other recent studies, according the specific characteristics of every adhesive. The authors discuss the findings of the paper about the characteristics and properties of adhesives for explain the results as polymerization, degradation of dentin-resin interface.

In this section the authors must report the limitations of the study.

Conclusions. This section is correct

Author Response

Introduction. This investigation is a paper that presents information for researchers and clinicians in the field of adhesives in restorative dentistry. Self-etch adhesives with one-step or two-steps were introduced to overcome the problems of etch and rinse systems and well accepted clinically due to their simple clinical application and time saving application. The objectives of this study were to evaluate the micro-tensile bond strength of several contemporary universal adhesives.

This section is correct.

  We appreciate reviewer for comments of Introduction.

Materials and methods. Globally, this section is correct with several subsections (Micro tensile bond strength test; Measurement of in situ degree of conversion (DC%), Statistical analysis) that showed the steps of the experiment. Each subsection must be report a number (2.1.; 2.2; …) Also, this section must report update references of each procedure.

We added numbers (2.1., 2.2., …) to each subsection, and added some references for each step in these subsections.

Results. This section is correct. The results are related with the specific characteristics of every adhesive. There are diferences  (i.e. tensile bond strength) in the different adhesives.

We appreciate reviewer for comments of Introdaction .            

  We appreciate reviewer for comments of Results.

Discussion. This section analyzes the most important results of the study and be compared with other recent studies, according the specific characteristics of every adhesive. The authors discuss the findings of the paper about the characteristics and properties of adhesives for explain the results as polymerization, degradation of dentin-resin interface.

In this section the authors must report the limitations of the study.

We have added content about the limitations of the study to the main text.

In added to,

 “The findings of the current study suggest that lately introduced so-called “univer-sal adhesives” demonstrate comparable bonding performance to dentin with gold-standard 2-SEA SE bond 2, however the more clinical bonding performance like simulated pulpal pressure or in situ clinical evaluation should be carried out in the fu-ture studies.”

Conclusions. This section is correct

  We appreciate reviewer for comments of Conclusions.

Round 2

Reviewer 1 Report

Comments and Suggestions for Authors

The authors responsed to all my comments

i can accept the paper now

Author Response

Thank you for reviewing.

I have checked the input form, but it seems that the download was not possible as you pointed out.

I apologize for the inconvenience, but could you please try submitting the form again? I

appreciate your cooperation.